# Novel Whey Fermented Beverage Enriched with a Mixture of Juice Concentrates: Evaluation of Antimicrobial, Antioxidant, and Angiotensin I Converting Enzyme Inhibitory (ACE) Activities Before and After Simulated Gastrointestinal Digestion

**DOI:** 10.3390/microorganisms13071490

**Published:** 2025-06-26

**Authors:** Paschalia Kotsaki, Maria Aspri, Photis Papademas

**Affiliations:** Department of Agricultural Sciences, Biotechnology and Food Science, Cyprus University of Technology, Limassol 3036, Cyprus; lina.kotsaki@cut.ac.cy (P.K.); maria.aspri@cut.ac.cy (M.A.)

**Keywords:** whey, fermentation, bioactive properties, sustainability, antimicrobial activity, antioxidant activity, gastrointestinal digestion, functional beverage, whey protein isolate, inulin

## Abstract

This study explored the development of a novel whey-based fermented beverage enriched with juice concentrates and health-promoting ingredients, emphasizing its bioactive properties. The formulation included whey protein isolate (5%), juice concentrates (10% apple, raspberry, and cranberry), and inulin (4%). Fermentation was carried out with the following strains: *Lacticaseibacillus rhamnosus* (LGG), *Lacticaseibacillus casei* (431), and *Lactobacillus helveticus* (R0052) at 2%. Antimicrobial activity was evaluated against pathogens including *Listeria monocytogenes* (strains 33423 and 33413), *Staphylococcus aureus* (113 and Newman), *Bacillus cereus* (DPC 6089), *Escherichia coli* (NCTC 9001), and *Salmonella Enteritidis* (NCTC 6676). Antioxidant capacity was measured using 2,2-Diphenyl-1-picrylhydrazylradical (DPPH) and Ferric Reducing Antioxidant Power (FRAP) assays, and angiotensin-converting enzyme (ACE) inhibitory activity was assessed. All bioactivities were found to be high in fermented whey beverage and a further significant increase was observed after simulated gastrointestinal digestion. This fruit-flavored whey beverage demonstrated notable antimicrobial and antioxidant activities, highlighting its potential for functional food applications aimed at combating harmful bacteria and oxidative stress.

## 1. Introduction

Approximately 3000 years ago, whey was first discovered as a by-product of cheesemaking since it is extracted from curd during processing [1,2]. It is a yellow–green liquid with a sour, slightly salty aftertaste. The yellow–green color of whey can be attributed to the presence of riboflavin (Vitamin B2) [3]. Whey is classified into two categories based on the method of milk coagulation i.e., sweet and acid whey. Sweet whey is obtained as a result of enzymatic milk coagulation performed with the use of chymosin. This type of whey has a pH in the range of 6–7 and it is also known as sweet cheese whey. Acid whey (pH 4.6–5.0) could occur as a filtrate from yogurt production (filtration is used to increase total solids) or from curd draining of cheeses left to drain overnight (i.e., Feta PDO) [4].

World whey output is about 180 million tons, containing some 1.5 million tons of high-value protein and 8.6 million tons of lactose [4]. Each year, the total production of whey increases by 1–2%, but less than half of the whey produced is utilized or processed using various technologies and methods [5].

Globally, approximately 40% of whey is discarded, resulting in a significant loss of valuable nutrients, while according to Food and Agriculture Organization (FAO), about one-third of global food production is lost or wasted every year [6].

The considerable lactose content in milk whey makes it a valuable raw material from an industrial perspective, possessing notable potential for the development of fermented products [7].

Fermented beverages produced using one or more bacterial cultures are increasingly recognized as functional foods, owing to their documented health-promoting properties. They constitute a significant component of the human diet and represent a primary source of beneficial microorganisms in daily consumption [8].

The use of lactic acid bacteria (LAB) in whey fermentation involves a vigorous bacterial metabolic process that affects carbohydrates, lipids, proteins, and allergenic peptides present in whey. Consequently, LAB enhances digestibility and contributes to the preservation of the whey [9]. Furthermore, the activity of LAB leads to an increase in the concentration of other metabolites, including aromatic compounds. These compounds significantly contribute to the overall flavor, texture, and sweetness of the final product [10].

The sensory acceptability of pure sweet whey is low due to its high mineral salt content, which results in an unpleasant flavor i.e., watery cheesy taste and aroma [11]. Currently, whey-based fruit beverages represent a viable option for utilizing whey in combination with a wide range of fruits [12]. A significant number of literature reviews focus on whey beverages, with a majority of them exploring the addition of probiotics, fruit juices, and vegetables [13,14,15,16,17,18,19,20].

Prebiotics are non-digestible food components that are selectively utilized by host microorganisms resulting in additional benefits to host health [21,22]. Among prebiotics, inulin is a soluble and fermentable fiber, and it is mostly used to obtain products with a low-fat content as it is usually used as a fat replacer, a sugar replacer, and a texture modifier, improving the rheological properties of the products [23,24]. The technological utilization of whey was reviewed, suggesting that the exploitation of whey results in the reduction in the environmental impact and leads to sustainable solutions practices by the dairy industry [25].

This study aimed to develop a whey-based, fruit-flavored beverage enriched with probiotics and a prebiotic, and to evaluate its antimicrobial, antioxidant, and angiotensin-converting enzyme (ACE) inhibitory activities both immediately after production and throughout refrigerated storage (4 ± 1 °C) for up to 28 days. The bioconversion of whey into a fruit-flavored functional beverage offers a promising and sustainable approach for the valorization of this dairy industry by-product.

## 2. Materials and Methods

### 2.1. Chemicals and Reagents

All chemicals, reagents, and organic solvents were purchased from Sigma-Aldrich (St. Luis, MO, USA) unless otherwise stated. Hippuryl-l-Histidyl-l-Leucine (HHL), 2, 2-Diphenyl-1-picrylhydrazyl radical (DPPH), FRAP (ferric reducing antioxidant powder) (Sigma-Aldrich Chemical Co., St. Louis, MO, USA), 2,4,6-tris (2-pyridyl)-S-triazine (TPTZ) (Sigma-Aldrich Chemical Co., St. Louis, MO, USA), were obtained from Sigma-Aldrich Chemical Co. (St. Louis, MO, USA). All other chemicals and reagents were of analytical grade.

### 2.2. Samples and Other Ingredients

Sweet whey (Charalambides Christis Ltd., Limassol, Cyprus), Fibruline Instant (COSUCRA Groupe Warcoing SA, Belgium, Brussels), Whey Protein Isolate (Alinda Velco s.a., Athens, Greece), Concentrated Fruit Juices (KEAN Soft Drinks LTD, Limassol, Cyprus).

### 2.3. Microbial Strains and Growth Conditions

The bacterial strains used in this study and their growth conditions are listed in Table 1. The selection of the LAB strains was based on their technological, probiotic, and safety profile. Pure *Lacticaseibacillus rhamnosus* (LGG), *Lacticaseibacillus casei* (431), and *Lactobacillus helveticus* (R0052) strains were obtained from Chr. Hansen A/S, Hørsholm, Denmark while stored at −80 °C until further analysis. All strains were maintained as frozen stock at −80 °C in De Man–Rogosa–Sharpe (MRS) or Brain Heart Infusion (BHI) broth supplemented with 20% glycerol. Prior to the experimental use, the cultures were propagated twice in MRS or BHI medium and incubated at 37 °C for 24 h.

### 2.4. Preparation of Whey Fermented Beverages

Whey samples were collected from a local dairy in Limassol, Cyprus (Charalambides Christis Ltd.), immediately after the production of Halloumi cheese. Halloumi is the traditional cheese of Cyprus that is produced using mixtures of cow, sheep, and goat milk.

Our sampling procedure was based on the discarded whey (a by-product of Halloumi cheese production) and was a mixture of different milk types (mixed whey after the production of Anari cheese with cow milk as a major ingredient with the addition of goat and sheep milk). Sampling was conducted in the morning and a second batch was collected in the afternoon of the same day. The whey samples were collected in sterilized 500 mL containers, placed in cool boxes, labeled, and immediately transported to the laboratory. They were kept at −80 °C for further analysis.

The whey was strained and heated to 60 °C for 5 min. This heating step is intended to improve the substrate’s ability to blend smoothly with the additional ingredients. After heating, WPI (5%) and fruit juice (10%) were added to obtain a thick, liquid-paste consistency. Finally, the whey mixture was homogenized by Ultraturrax (IKA^®^—Werke GmbH & Co. KG, Mindelheim, Germany) to create a homogeneous, smooth mixture with a thicker texture and improved mixability. Then, it was pasteurized at 72 °C for 15 s and cooled to 30–40 °C. In the pasteurized whey 4% (*w*/*v*) of inulin was inoculated with 2% (*v*/*v*) probiotic bacteria and incubated at 37 °C for 18 h (based on optimum growth) (see Figure 1).

The final whey samples were stored at 4 °C until further analysis was performed. Finally, 8 samples of fermented whey from two different batches with each of the previous 3 selected LAB strains were prepared (Table 2).

### 2.5. Nutritional Analysis of the Whey and the Probiotic Whey Drink with Added Juice Concentrates

The whey chemical composition (fat, protein, and total solids) was determined by the spectrophotometric method in infrared with a MilkoScan FT 120 apparatus (FossElectric, Hillerød, Denmark). The chemical composition (fat, protein, ash, and total solids) of the probiotic drink with the addition of juice concentrates was evaluated according to Association of Official Analytical Chemists (AOAC) (2016). For the determination of mineral content, it was performed microwave digestion with inductively coupled plasma optical emission spectroscopy (ICP-OES) according to AOAC (2016). The following micronutrients were analyzed: sodium, potassium, phosphorus, and calcium. Results were expressed in mg/L.

The pH of whey and the fermented whey beverage was directly measured with a pH-meter (Hanna, Woonsocket, RI, USA).

### 2.6. Probiotic Viability

The viability of probiotics in whey-fermented beverages was assessed after 1, 14, and 28 days of refrigerated storage. The samples were submitted to serial decimal dilutions in peptone water and pour-plated in deMan–Rogosa–Sharpe agar (MRS Agar, Merck, Darmstadt, Germany) followed by anaerobic incubation at 37 °C for 48 h. Anaerobic conditions were maintained using an anaerobic jar (Merck) with an Anaerocult^®^ A kit (Merck). The results were expressed as the logarithm of colony-forming units per milliliter of product (log CFU mL⁻¹). All analyses were conducted in duplicate.

### 2.7. In Vitro Digestion of Whey Fermented Beverages

Samples were subjected to in vitro digestion using a model based on human gastrointestinal physiologically relevant conditions (oral, gastric, and intestinal phases). In vitro digestion was carried out following the INFOGEST protocol [26,27].

The simulated digestion fluids were prepared in advance and frozen (−20 °C) until needed. On the day of the experiment, the solutions were thawed and both CaCl_2_ and the required enzymes were prepared before use.

Briefly, the oral phase was initiated by the addition of 4 mL of pre-warmed simulated salivary fluid (pH 7.0) to 5 mL of each whey sample. Then, 6.25 µL of 0.3 M CaCl_2_ (H_2_O)_2_, 243.5 µL of water, and NaOH (up to 1 mL) were added and thoroughly mixed to adjust the pH to 7. The oral bolus was incubated with rotation at 30 rpm for 2 min at 37 °C.

For the gastric phase, the resulting solution was diluted with 8 mL of simulated gastric fluid at 37 °C, 1 mL of porcine pepsin solution at the final activity of 2000 U/mL and 5 μL of CaCl_2_ (300 mM) followed by 1.25 μL of 0.3 M CaCl_2_. The pH was adjusted to 3.0 using HCl (1 M). The gastric bolus was incubated for 2 h at 37 °C under rotation.

For the intestinal phase, the gastric chyme was mixed with 8 mL of simulated intestinal fluid (SIF), 3 mL of bile solution (prepared in SIF, 10 mM digested) and 5 mL of pancreatin (prepared in SIF to achieve a trypsin activity of 100 U/mL digested), and 40 μL of CaCl_2_ (300 mM) were added to the 20 mL of the gastric chyme and filled up to 40 mL with dH_2_O and NaOH. The pH was verified and adjusted to 7.0 when necessary, followed by incubation for 120 min at 37 °C. After the completion of the intestinal phase, digested samples were immediately heated at 85 °C for 10 min and placed on ice to deactivate enzymatic activities. Samples were stored at −18 °C until analyzed. In vitro digestion was carried out in duplicates for each sample.

### 2.8. Antimicrobial Activity of Whey Fermented Beverages

Prepared fermented whey-based drinks were screened for their antimicrobial activity against various indicator organisms using an agar well diffusion assay. The antagonistic activity of test cultures and whey beverage was assessed against common foodborne pathogens such as *Listeria monocytogenes Staphylococcus aureus*, *Bacillus cereus*, *Escherichia coli*, and *Salmonella enteritidis* (Table 2).

A well diffusion assay was performed with some modifications [27]. A solution of agar was heated until it reached a molten state and then cooled down to a temperature of 48 °C. Following that, a 50 μL culture of the indicator strain, which had been grown in the suitable medium to the early stationary phase, was introduced into 20 mL of media. The plates were allowed to solidify and dry before making wells (6 mm in diameter) in the seeded plates. Aliquots of (50 μL) fermented whey-based drinks were then carefully dispensed into the wells of the plates. The plates were subsequently incubated at the appropriate temperature according to Table 2. The diameter of the zone of inhibition extending laterally around the well was calculated as the diameter of the zone of clearing minus the diameter of the well (6 mm), using an electronic caliper with a digital display.

### 2.9. Antioxidant Activity Assays of Whey Fermented Beverages

#### 2.9.1. DPPH (2, 2 Diphenyl-1-Picryl Hydrazyl) Assay

The antioxidant activity of whey samples before and after gastrointestinal digestion was determined by the ability of the samples to scavenge 2,2-diphenyl-1-picrylhydrazyl (DPPH) radical with a spectrophotometric assay according to Brand-Williams, Cuvelier, and Berset (1995) with some modifications [28].

In brief, 0.1 mL samples were transferred to test tubes with 3.9 mL radical DPPH (60 mol/L DPPH solution) and homogenized and incubated for 30 min in the dark at room temperature. The absorbance of each sample was determined at 517 nm. All determinations were carried out in duplicate. Methanol was used as a blank, while the methanolic DPPH solution was used as a control. The DPPH activity was calculated using the Equation:(1)$$DPPH\left(\%\right)=\left(\frac{A\;control-A\;sample}{A\;control}\right)\times 100$$
where A control is the absorbance of the control and A sample is the absorbance of the sample. Results were expressed as millimoles of Trolox equivalent antioxidant capacity (TEAC) per 100 mL of sample (TEAC mmoles/100 mL).

#### 2.9.2. Ferric Reducing Antioxidant Power (FRAP) Assay

The antioxidant activity of the whey samples was measured using the ferric reducing antioxidant power (FRAP) assay [29,30].

The freshly made-up FRAP solution contained TPTZ (10 mM, in 40 Mm hydrochloric acid), ferric chloride (20 Mm), and acetate buffer (0.3 M, pH 3.6) by a ratio of 1:1:10 (*v*/*v*/*v*). This solution was used as blank. A total of 1 mL of sample solutions at different concentrations were added to 5 mL of FRAP reagent. Following that, the mixture was incubated at 37 °C for 20 min.

The ferric-reducing ability of byproduct extracts was measured by monitoring the increase in absorbance at 593 nm for 45 min. Standard solutions of ferrous sulfate (100–1400 μM) were used for the calibration curve.

Results were expressed as FRAP value (μM Fe^2+^), standard curve equation was:(2)$$y=0.0023x+0.1399\;(R^{2}=0.9995)$$

The higher the FRAP value, the greater the ferric-reducing antioxidant capacity.

### 2.10. Angiotensin-I-Converting Enzyme (ACE)-Inhibitory Activity Assay

The ACE-inhibitory activity was determined based on the method described with some modifications and was assessed spectrophotometrically [30]. Briefly, a solution of hippuryl–histidyl–leucine (HHL, 5 mM) was prepared in sodium phosphate buffer (NaPB, 0.1 M, pH 8.3) containing NaCl (0.3 M). ACE (Angiotensin Converting Enzyme from rabbit lung, 1 unit/mL, EMD Millipore) was prepared (1 U/mL) with potassium phosphate buffer (KPB, 0.01 M, pH 7) containing NaCl (500 mM) and the mixture was preincubated at 37 °C for 5 min. The reaction was initiated by the addition of 10 µL of ACE solution (100 mU/mL), and the mixture was incubated at 37 °C for 30 min. For each analysis, 40 µL of sample, 100 µL of HHL, and 10 µL of ACE solutions were added in an Eppendorf. Samples were incubated for 60 min at 37 °C, and then 125 µL of HCl (1N) was added to stop the reaction.

The absorbance was measured at 420 nm using the Tecan Infinite 200 PRO microplate plate reader. The assay was performed by using the same samples in duplicate with two wells per sample. The percentage of ACE inhibition (ACEi%) was calculated by subtracting the Hippuric Acid (HA) produced in the presence of the inhibitors from the HA produced in the absence of inhibitors (under the same conditions) as shown below:(3)$$ACEi\%=100\times \frac{\left({HA}_{control}-{HA}_{sample}\right)}{{HA}_{control}}$$
where ACEi% is the inhibition percentage of ACE, HA_Sample_ is the concentration of the released HA (in the presence of inhibitor) and HA_Control_ is the concentration of HA in the control blank (without inhibitor). 

### 2.11. Statistical Analysis

All experiments were carried out in duplicate. Data are expressed as the mean ± standard deviation (SD). Statistical analysis was performed using SPSS version 17.0 (SPSS Inc., Chicago, IL, USA). Univariate Analysis of Variance (ANOVA) was performed employing post-hoc Duncan’s test. Values were considered to be significantly different from each other at a significance level of *p* < 0.05.

## 3. Results and Discussion

### 3.1. Nutritional Analysis of the Whey and the Probiotic Whey Drink with Added Juice Concentrates

Whey, a valuable byproduct of cheese production, is rich in essential nutrients, including proteins, lactose, vitamins, and minerals. However, its low protein content and slightly acidic nature limit its direct consumption. The physicochemical properties of the whey used as a raw material were as follows: pH of 6.21 ± 0.14, 5.28 ± 0.34 g/L of total solids, 0.76 ± 0.08 g/L of total proteins, and 0.02 ± 0.02 g/L of fat. In the present study, the composition and pH of the whey showed similar values to those presented by Tamime [31]. According to this author, sweet whey typically shows a pH from 5.9 to 6.6 and around 6.5% solids, which includes 4.8% lactose, 0.9% protein, 0.1% lipids, and 0.7% minerals.

To enhance its nutritional profile and functionality, whey can be incorporated into probiotic beverages with added ingredients such as whey protein isolate (WPI), inulin, and fruit juice concentrates. Table 3 presents the physicochemical characterization of fermented probiotic whey drink with added juice concentrates. The formulated probiotic whey beverage in this study contains 3.89% ± 0.11 protein, 0.09% ± 0.01 fat, 1.18 ± 0.19 ash, and a total solid content of 21.35% ± 0.35. The addition of WPI (5%) significantly boosts its protein content, making it a rich source of essential amino acids necessary for muscle repair, immune function, and overall metabolic health. Inulin (4%), a prebiotic fiber, enhances the growth and activity of beneficial gut bacteria, improving digestion and gut health. Additionally, fruit juice concentrates (5% apple, 2.5% raspberry, and 2.5% cranberry) could contribute to the beverage’s antioxidant properties, flavor profile, and vitamin content.

Whey is rich in essential micronutrients such as sodium, potassium, calcium, and phosphorus, all of which play a vital role in supporting the functionality of whey proteins [25]. The mineral content of the fermented probiotic whey drink with added juice concentrates is presented in Table 3. Whey drink is an excellent source of bioavailable calcium (511.80 mg/L ± 53.02) that improves bone health and reduces the risk of high blood pressure [32]. Moreover, calcium derived from whey is easily absorbed in the intestine due to the presence of lactose. In addition to providing bone and teeth strength, phosphorus is extremely effective in performing critical functions for many human organs and is essential for energy metabolism and cellular function. Potassium (2039.69 mg/L) helps regulate blood pressure, nerve signaling, and muscle contractions, whereas sodium (2943.03 mg/L) is necessary for maintaining fluid balance and nerve transmission [33]. Results showed that in the fermented whey beverage, calcium, phosphorus, potassium, and sodium contributions to DRI were 10.2%, 12.5%, 8.7%, and 39.3%, respectively, in adults if 200 mL of the beverage is taken daily. The pH of 3.65 in the fermented whey drink contributes to its refreshing taste.

### 3.2. Probiotic Viability

The fermented whey beverages were evaluated on days 1, 14, and 28 of refrigerated storage (4 ± 1 °C) to assess the viability of potential probiotic strains. The viable count of probiotic bacteria of whey fermented beverages during storage is presented in Figure 2. A key indicator of probiotic product quality is the count of viable probiotic bacteria, which should remain at or above 6 log CFU/mL until expiration. After the first day of storage at 4 °C, the probiotic counts in fermented whey beverages without juice concentrate were as follows: *Lactobacillus rhamnosus* (LGG) at 7.90 ± 0.06 log CFU/mL, *Lacticaseibacillus casei* 431 at 7.99 ± 0.01 log CFU/mL, and *Lactobacillus helveticus* (R0052) at 7.76 ± 0.05 log CFU/mL. Similar results were observed in fermented whey beverages containing juice concentrate, showing that the survival kinetics of probiotic strains in the fermented whey beverages were not affected by the addition of juice concentrate as compared with the no juice-added whey beverage. Probiotic counts remained stable for all fermented whey beverages for up to 14 days, except for sample 4ND, which showed a significant decrease to 7.02 ± 0.03 log CFU/mL. By the end of the storage period, probiotic viability had moderately declined but remained above the recommended threshold of 10^6^ CFU/mL. This finding is particularly important, as it ensures that the formulated dairy beverages provide a sufficient daily intake of viable probiotic cells, confirming their functional potential.

### 3.3. Antimicrobial Activity

The antimicrobial activity of fermented whey-fruit flavored beverages after in vitro digestion was evaluated against six foodborne pathogens on Days 1, 14, and 28. Results are presented in Table 4. The tested samples were active against both Gram-positive and Gram-negative pathogens. The zone of inhibition ranged from 16.13 ± 0.11 (sample 1D day 28) to 29.46 ± 0.06 (Sample 4D day 1). The inhibitory spectrum of the strains showed strong activity against *Listeria* spp. in digested samples. Maximum zone of inhibition was observed against *L. monocytogenes* 33423, followed by *S. aureus Newman*, *Salmonella enteritidis* NCTC 6676, *S. aureus* 113, *B. cereus* DPC 6089, and *L. monocytogenes* 33413 in digested samples. For *Listeria monocytogenes* 33423, the highest activity was observed on Day 1, with Sample 4D exhibiting the greatest inhibition (29.46 ± 0.06 mm), followed by a gradual decline over time. Similarly, *Listeria monocytogenes* 33413 demonstrated consistent activity across all time points, with Sample 7D showing slightly higher inhibition (21.35 ± 0.08 mm on Day 1 and 20.79 ± 0.04 mm on Day 14). Activity against *Staphylococcus aureus* 113 was the highest antimicrobial activity observed on Day 1 with Sample 6D (23.03 ± 0.04 mm) but showed a marked decline by Day 28. For *Staphylococcus aureus* Newman, Sample 6D consistently exhibited superior inhibition, with the highest activity recorded on Day 1 (26.59 ± 0.11 mm), followed by a gradual reduction. Against *Bacillus cereus* DPC 6089, moderate inhibition was noted across all samples, with peak activity observed in Sample 4D on Day 1 (22.45 ± 0.05 mm), declining steadily over time. In contrast, *Salmonella* Enteritidis NCTC 6676 maintained stable activity across all time points, with Sample 6D showing the highest inhibition on Day 14 (24.08 ± 0.08 mm). Overall, Sample 6D demonstrated high antimicrobial activity across multiple foodborne pathogens used in this study, while a general trend of reduced antimicrobial inhibition was observed from Day 1 to Day 28.

The antibacterial activity in fermented whey beverages arises from the synergistic effects of organic acids, bioactive peptides, bacteriocins, and other antimicrobial compounds produced during fermentation. This combination not only enhances the safety and shelf-life of these beverages but also provides potential health benefits.

Studies have demonstrated that fermented whey beverages show strong antibacterial activity, particularly against pathogens associated with foodborne illnesses. For instance, one study found that a whey beverage fermented with *Lactobacillus rhamnosus* inhibited *E. coli* and *Salmonella typhimurium*, suggesting potential for food preservation and health applications [34].

### 3.4. Antioxidant Activity

Antioxidant activity can be assessed through various mechanisms and reactions, including radical scavenging, reducing power, inhibition of chain initiation, binding to transition metal ion catalysts, and prevention of lipid peroxidation. Therefore, it is essential to use multiple methods to evaluate antioxidant activity. In this study, the antioxidant activity of fermented whey-fruit flavored beverages before and after the samples were subjected to gastrointestinal digestion was analyzed using DPPH and FRAP assays. Results of the antioxidant activity of whey-fruit flavored beverages during refrigerated storage are presented in Figure 3 and Figure 4.

Based on DPPH radical scavenging activity, the antioxidant activity of unfermented non-digested whey samples (ND1 and ND8) was 10.65% ± 0.32 on Day 1 and 9.62% ± 0.02 on Day 28 for sample ND1, and for sample ND8 was 16.02% ± 0.10 in Day 1 and 15.18% ± 0.12 in Day 28.

After fermentation, the antioxidant activity of the fermented whey samples was increased which is in agreement with other studies [16,33,35]. In non-digested whey samples, Sample 5ND Day 1 showed the highest percentage of DPPH radical inhibition (43.26% ± 0.08). In juice whey samples, Sample 6ND Day 28 exhibited DPPH activity of 41.07% ± 0.20. The increase in DPPH radical scavenging in fermented beverages is probably a result of the ability of probiotics to enhance metabolic antioxidant capacity. This may be attributed to the production of metabolites with higher and better antioxidant activity like glutathione, and butyrate or to the chelation of metal ions. One of the potential factors could be the release of bio-accessible phenolic compounds [33].

The further increase in the antioxidant activity for all the samples after in vitro gastrointestinal digestion is in agreement with other studies that demonstrated that the antioxidant properties of whey proteins could be increased through hydrolysis [36]. Also, the addition of juice improved both DPPH and FRAP radical scavenging activity which is in line with other studies [16]. From Day 1 to Day 28 of storage at 5 °C, the DPPH and FRAP scavenging activity of whey samples gradually decreased. In particular, in digested whey samples, Sample 5D Day 1 showed the highest percentage of DPPH radical inhibition (74.83% ± 0.02) while Sample 1D Day 28 showed the lowest percentage of DPPH radical inhibition (14.09% ± 0.02). In juice whey samples, Sample 6D Day 28 showed the lowest percentage of DPPH radical inhibition (67.05% ± 0.21).

FRAP activity could be related to phenolic content which is involved in the reduction of 2, 4, 6-tris (2-pyridyl)-S-triazine TPTZ–Fe^3+^ complex to TPTZ–Fe^2+^ form [37].

In non-digested whey samples, sample 6ND Day 1 showed the highest percentage of FRAP radical inhibition (49.29 μmolTEs/g protein ± 0.57) while sample 1ND Day 28 showed the lowest percentage of FRAP radical inhibition (12.05 μmolTEs/g protein ± 1.32). In juice whey samples, sample 5ND Day 28 showed the lowest percentage of FRAP radical inhibition (33.19 μmolTEs/g protein ± 0.88).

Digested whey samples also demonstrated enhanced FRAP values compared to non-digested whey samples. Sample 7D Day 1 showed a strong antioxidant activity of FRAP radical inhibition (66.53 μmolTEs/g protein ± 1.22) while Sample 1D Day 28 showed the lowest percentage of FRAP radical inhibition (17.92 μmolTEs/g protein ± 0.66). In juice whey samples, Sample 5D Day 28 showed the lowest percentage of FRAP radical inhibition (49.63 μmolTEs/g protein ± 2.81).

### 3.5. ACE Inhibitory Activity

Hypertension is now recognized as one of the leading risk factors for the development of cardiovascular diseases. Alongside maintaining a healthy lifestyle and diet, consuming fermented dairy products enriched with bioactive peptides may play a crucial role in hypertension prevention. ACE (angiotensin-converting enzyme; EC 3.4.15.1) is a multifunctional ectoenzyme found in various tissues, serving critical physiological functions in the renin–angiotensin, kallikrein–kinin, and immune systems [38]. This enzyme contributes to increased blood pressure by converting the inactive angiotensin-I into the potent vasoconstrictor angiotensin-II and degrading bradykinin, a key vasodilator [39]. Consequently, inhibiting ACE is an effective strategy for preventing and managing high blood pressure.

Changes in ACE inhibitory activity (%) of whey-fruit flavored beverages during refrigerated storage before and after in vitro digestion are shown in Figure 5. According to the results obtained from the ACE inhibitory activity, a gradual decrease in ACE inhibition was observed for both cases of digested and non-digested samples. In non-digested whey samples, Sample 7ND Day 1 showed the highest percentage of ACE inhibitory activity (67.19% ± 0.75) while Sample 1ND Day 28 showed the lowest percentage of ACE inhibitory activity (11.24% ± 0.70). In juice whey samples, Sample 6ND Day 28 exhibited the lowest ACE inhibitory activity of 50.74% ± 3.67.

In digested whey samples, Sample 6D Day 1 showed the highest percentage of ACE inhibitory activity (78.63% ± 0.40) while Sample 1D Day 28 showed the lowest percentage of ACE inhibitory activity (15.58% ± 0.57). In juice whey samples, Sample 5D Day 28 showed the lowest percentage of ACE inhibitory activity (67.09% ± 2.69).

## 4. Conclusions

Functional beverages offer health-oriented, nutritious options enriched with bioactive compounds that support well-being and specific health outcomes. Moreover, their production can enhance environmental sustainability by utilizing dairy by-products like whey, reducing food waste, and improving resource efficiency.

This study focused on a whey-based beverage fermented with lactic acid bacteria and enriched with concentrated fruit juice for improved flavor and additional health benefits; the beverage demonstrated notable antimicrobial and antioxidant activities. These findings suggest its potential to combat harmful bacteria, reduce oxidative stress, and support immune function.

Additionally, the incorporation of sweet whey—a by-product of cheese production—into the beverage formulation underscores the product’s alignment with sustainable food processing practices. Transforming whey into a functional beverage not only contributes to human health but also supports waste reduction and improves resource efficiency within the dairy industry. The resulting fruit-flavored fermented whey beverage thus represents a novel and environmentally sustainable addition to the functional beverage sector.

Sensory analysis was not conducted, as the primary focus of this study was the assessment of the beverage’s functional properties and its shelf-life stability during refrigerated storage. Given the importance of consumer acceptance for successful product development, further studies are needed to evaluate its sensory properties in terms of flavor, aroma, and texture.

## Figures and Tables

**Figure 1 microorganisms-13-01490-f001:**
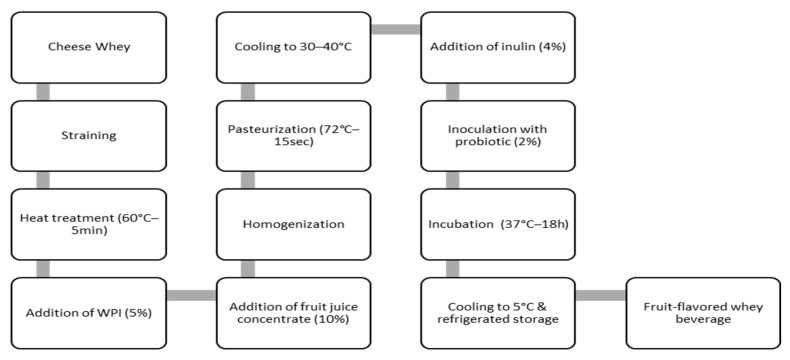
Production process of fermented whey beverages.

**Figure 2 microorganisms-13-01490-f002:**
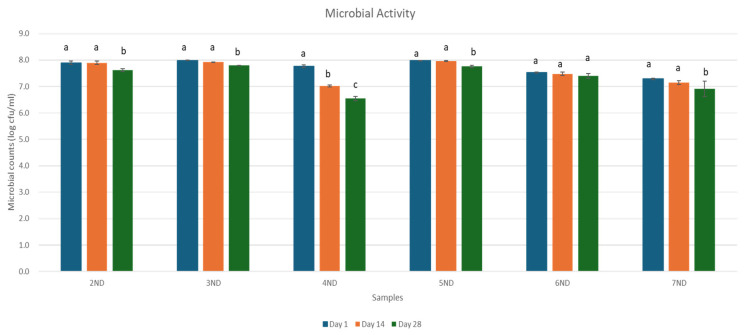
Viability of probiotic strains in different fermented whey beverages during storage at 4 °C. Lowercase letters indicate significant differences (*p* < 0.05) between the storage days of each fermented whey beverage sample. Error bars represent the mean (n = 2) ± standard deviation (SD). ND = non-digested samples.

**Figure 3 microorganisms-13-01490-f003:**
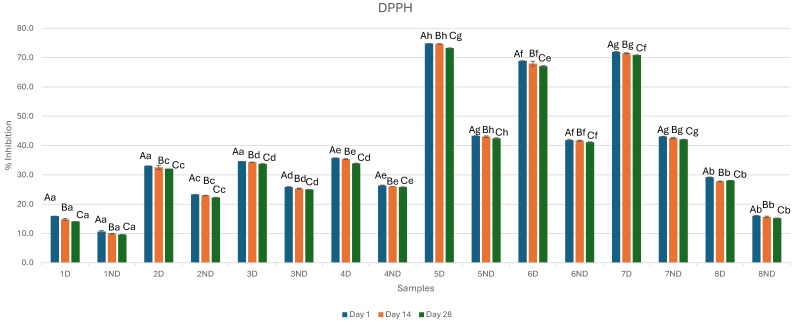
Effect of fermentation, digestion, and addition of juice concentrate on DPPH radical scavenging activity on digested samples during 28 days of storage (measurements were obtained on Day 1, 14, and 28). Uppercase letters represent differences between days, while lowercase letters indicate statistically significant differences within the same column. Specifically, means in the same column that are marked with different lowercase letters are significantly different at a level of *p* < 0.05. Sample 1D = Digested Sample 1, Sample 1ND = Non-Digested Sample 1.

**Figure 4 microorganisms-13-01490-f004:**
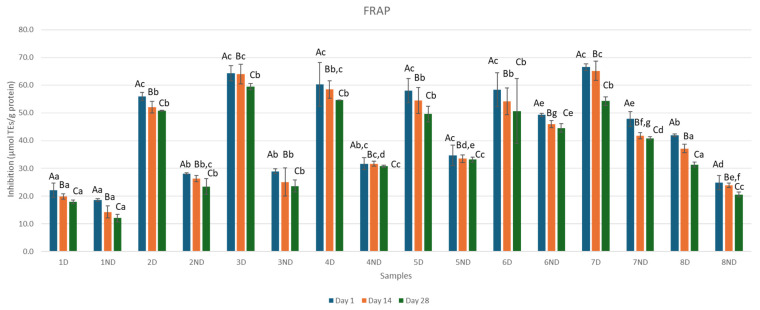
Effect of fermentation, digestion, and addition of juice concentrate on FRAP ferric reducing antioxidant power on digested samples during 28 days of storage (measurements were obtained on Day 1, 14, and 28). Uppercase letters represent differences between days, while lowercase letters indicate statistically significant differences within the same column. Specifically, means in the same column that are marked with different lowercase letters are significantly different at a level of *p* < 0.05. Sample 1D = Digested Sample 1, Sample 1ND = Non-Digested Sample 1.

**Figure 5 microorganisms-13-01490-f005:**
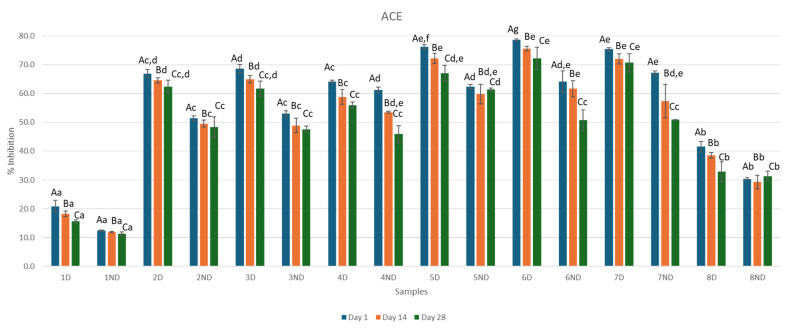
Effect of fermentation, digestion, and addition of juice concentrate on ACE inhibitory activity on digested samples during 28 days of storage (measurements were obtained on Day 1, 14, and 28). Uppercase letters represent differences between days, while lowercase letters indicate statistically significant differences within the same column. Specifically, means in the same column that are marked with different lowercase letters are significantly different at a level of *p* < 0.05. Sample 1D = Digested Sample 1, Sample 1ND = Non-Digested Sample 1.

**Table 1 microorganisms-13-01490-t001:** Bacterial strains used in the study.

Bacterial Strain	Origin/Characteristics	Microbiological Media	Incubation Conditions
	Indicator Strains		
*Listeria monocytogenes* 33423	UCC Culture Collection/Food Isolate	BHI	37 °C/48 h
*Listeria monocytogenes* 33413	UCC Culture Collection/Food Isolate United Kingdom outbreak (*L. mono* Ts45)	BHI	37 °C/48 h
*Staphylococcus aureus* SA 113	UCC Culture Collection/Representative staphylococcal organism in model virulence studies	BHI	37 °C/48 h
*Staphylococcus aureus* Newman	UCC Culture Collection	BHI	37 °C/48 h
*Bacillus cereus* DPC 6089	DPC culture Collection	BHI	37 °C/48 h
*Escherichia coli* NCTC 9001	NCTC Culture Collection	BHI	37 °C/48 h
*Salmonella* Enteritidis NCTC 6676	NCTC Culture Collection	BHI	37 °C/48 h
	LAB used for the production of fermented whey beverages		
*Lacticaseibacillus rhamnosus* (LGG)		MRS	37 °C/24 h
*Lacticaseibacillus casei* (431)		MRS	37 °C/24 h
*Lactobacillus helveticus* (R0052)		MRS	37 °C/24 h

**Table 2 microorganisms-13-01490-t002:** Whey samples used in this study.

Sample Number	Probiotic	Inulin	Whey Protein Isolate (WPI)	Juice Concentrate *^1^
1	No	No	No	No
2	*Lactobacillus rhamnosus* (LGG)—2%	4%	5%	No
3	*Lacticaseibacillus casei (L. casei)*—2%	4%	5%	No
4	*Lactobacillus helveticus* (R0052) 2%	4%	5%	No
5	*Lactobacillus rhamnosus* (LGG) 2%	4%	5%	10%
6	*Lacticaseibacillus casei (L. casei)* 2%	4%	5%	10%
7	*Lactobacillus helveticus* (R0052) 2%	4%	5%	10%
8	No	4%	5%	No

*^1^ addition of 5% Apple juice concentrate, 2.5% Raspberry juice concentrate, and 2.5% Cranberry juice concentrate.

**Table 3 microorganisms-13-01490-t003:** Gross chemical composition of the probiotic whey drink with added juice concentrates.

Parameter	Probiotic Whey Drink
Fat %	0.09 ± 0.01
Protein %	3.89 ± 0.11
Ash %	1.18 ± 0.19
Total Solids %	21.35 ± 0.35
Calcium mg/L	511.80 ± 53.02
Phosphorus mg/L	437.17 ± 69.08
Potassium mg/L	2039.69 ± 204.59
Sodium mg/L	2943.03 ± 683.96
pH	3.65

(Inulin 4%, WPI 5%, Juice (5% apple concentrate, 2.5% raspberry, 2.5% cranberry). Values are means ± SD of duplicates (n = 2).

**Table 4 microorganisms-13-01490-t004:** Antimicrobial activity of fermented whey-fruit flavored beverages after in vitro simulated gastrointestinal digestion (D = digested samples) *.

Indicator Strains	Days	Sample 1D	Sample 2D	Sample 3D	Sample 4D	Sample 5D	Sample 6D	Sample 7D	Sample 8D
*Listeria monocytogenes* 33423	1	27.92 ± 0.04 ^a^	28.14 ± 0.06 ^a.b^	28.57 ± 0.06 ^b^	29.46 ± 0.06 ^c^	28.32 ± 0.06 ^a.b^	29.20 ± 0.05 ^c^	28.45 ± 0.63 ^a.b^	28.49 ± 0.04 ^a.b^
	14	24.05 ± 0.05 ^a^	26.41 ± 0.05 ^c^	24.13 ± 0.04 ^a^	26.09 ± 0.04 ^b.^	26.36 ± 0.06 ^c^	27.86 ± 0.04 ^e^	27.62 ± 0.10 ^d^	27.52 ± 0.03 ^d^
	28	21.86 ± 0.04 ^a^	22.14 ± 0.04 ^b^	22.76 ± 0.06 ^c^	24.47 ± 0.03 ^f^	23.05 ± 0.05 ^d^	24.63 ± 0.05 ^g^	24.51 ± 0.04 ^f^	24.09 ± 0.05 ^e^
*Listeria monocytogenes* 33413	1	19.45 ± 0.06 ^a^	20.12 ± 0.06 ^b^	20.68 ± 0.08 ^c^	20.90 ± 0.08 ^d^	21.25 ± 0.06 ^e^	20.70 ± 0.11 ^c^	21.35 ± 0.08 ^e^	21.17 ± 0.08 ^e^
	14	19.34 ± 0.05 ^a^	19.50 ± 0.06 ^a.b^	19.9 ± 0.11 ^d.e^	19.74 ± 0.03 ^c.d^	19.80 ± 0.11 ^d^	20.06 ± 0.08 ^e^	20.79 ± 0.04 ^f^	19.6 ± 0.04 ^c.b^
	28	19.09 ± 0.07 ^a^	19.14 ± 0.08 ^a.b^	19.89 ± 0.04 ^e^	19.29 ± 0.09 ^b^	19.68 ± 0.10 ^d^	19.95 ± 0.04 ^e^	19.95 ± 0.04 ^e^	19.5 ± 0.10 ^c^
*Staphylococcus aureus* 113	1	21.30 ± 0.08 ^d^	19.73 ± 0.04 ^b^	19.57 ± 0.08 ^a^	20.55 ± 0.02 ^c^	21.84 ± 0.06 ^f^	23.03 ± 0.04 ^h^	22.38 ± 0.12 ^g^	21.52 ± 0.06 ^e^
	14	19.11 ± 0.06 ^a^	20.29 ± 0.03 ^d^	19.5 ± 0.06 ^a^	20.48 ± 0.06 ^e^	19.45 ± 0.05 ^b^	19.34 ± 0.06 ^b^	20.53 ± 0.04 ^e^	19.91 ± 0.04 ^c^
	28	16.13 ± 0.11 ^a^	18.32 ± 0.08 ^e^	17.06 ± 0.08 ^b^	17.8 ± 0.06 ^d^	19.20 ± 0.05 ^g^	17.27 ± 0.04 ^c^	18.78 ± 0.10 ^f^	16.25 ± 0.04 ^a^
*Staphylococcus aureus Newman*	1	24.04 ± 0.05 ^b^	24.36 ± 0.04 ^c^	24.36 ± 0.06 ^c^	23.59 ± 0.02 ^a^	24.69 ± 0.05 ^d^	26.59 ± 0.11 ^g^	25.10 ± 0.06 ^e^	26.39 ± 0.06 ^f^
	14	23.49 ± 0.07 ^a^	23.92 ± 0.04 ^c^	23.59 ± 0.07 ^a^	24.23 ± 0.05 ^d^	24.19 ± 0.05 ^d^	24.27 ± 0.05 ^d^	26.11 ± 0.05 ^e^	23.76 ± 0.08 ^b^
	28	21.1 ± 0.04 ^a^	21.64 ± 0.04 ^c^	21.24 ± 0.04 ^b^	22.41 ± 0.06 ^e^	22.15 ± 0.06 ^d^	22.47 ± 0.05 ^f^	22.80 ± 0.06 ^e^	22.16 ± 0.04 ^d^
*Bacillus cereus* DPC 6089	1	20.63 ± 0.04 ^a^	20.95 ± 0.04 ^b^	20.86 ± 0.04 ^b^	22.45 ± 0.05 ^e^	21.32 ± 0.10 ^c^	21.27 ± 0.04 ^c^	21.61 ± 0.10 ^d^	21.52 ± 0.05 ^d^
	14	19.92 ± 0.04 ^a^	20.79 ± 0.08 ^c^	20.26 ± 0.06 ^a^	21.82 ± 0.10 ^e^	21.13 ± 0.06 ^d^	21.05 ± 0.04 ^d^	22.40 ± 0.05 ^f^	21.09 ± 0.12 ^d^
	28	19.05 ± 0.06 ^a^	19.27 ± 0.08 ^b^	19.68 ± 0.09 ^c^	19.94 ± 0.04 ^d.e^	20.11 ± 0.10 ^e^	19.18 ± 0.11 ^c.d^	20.33 ± 0.06 ^f^	19.09 ± 0.06 ^a.b^
*Salmonella* Enteritidis NCTC 6676	1	23.47 ± 0.06 ^b^	22.8 ± 0.10 ^a^	23.83 ± 0.06 ^b^	23.77 ± 0.07 ^b^	24.18 ± 0.06 ^c^	24.42 ± 0.07 ^d^	23.16 ± 0.06 ^b^	23.87 ± 0.05 ^b^
	14	23.06 ± 0.08 ^b^	22.91 ± 0.04 ^a^	23.8 ± 0.06 ^f^	23.21 ± 0.06 ^c^	23.60 ± 0.04 ^e^	24.08 ± 0.08 ^g^	23.36 ± 0.06 ^d^	23.51 ± 0.06 ^e^
	28	22.57 ± 0.04 ^a^	22.77 ± 0.08 ^b^	23.17 ± 0.03 ^d.e^	22.60 ± 0.05 ^a^	22.91 ± 0.03 ^c^	23.24 ± 0.04 ^e.f^	23.34 ± 0.04 ^f^	23.06 ± 0.07 ^d^

* Values are the mean of duplicate zones of inhibition using well assays (mm) ± SD. Different letters in the same row indicate significant statistical differences (*p* < 0.05).

## Data Availability

The original contributions presented in this study are included in the article. Further inquiries can be directed to the corresponding author.

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
