# Peer review of "Novel Whey Fermented Beverage Enriched with a Mixture of Juice Concentrates: Evaluation of Antimicrobial, Antioxidant, and Angiotensin I Converting Enzyme Inhibitory (ACE) Activities Before and After Simulated Gastrointestinal Digestion"

_microorganisms, 2025, doi:10.3390/microorganisms13071490_

Round 1
Reviewer 1 Report
Comments and Suggestions for Authors
The manuscript is not well designed and the scientific issues were not well addressed.
1. What is "enhanced bioavailability of antioxidants", it is not clear
2. The significance of this study is not well stated
3."an acidic environment suitable for probiotic stability" is incorrect
4. section 2.1 is just basic nutritional information, not Physiochemical analysis
5. n=2 is unacceptable
6. no unit in table 2
Author Response
Reviewer 1
The manuscript is not well designed and the scientific issues were not well addressed.
- What is "enhanced bioavailability of antioxidants", it is not clear – the sentence was rephrased
- The significance of this study is not well stated. The significance of the study was stated at the Introduction and Conclusion sections. Specifically, we emphasize that this research contributes to the development of a sustainable, health-promoting functional beverage by utilizing whey—a major dairy by-product—thus addressing both nutritional innovation and environmental sustainability. Additionally, the demonstrated antimicrobial and antioxidant properties of the fermented whey beverage support its potential role in enhancing gut health and reducing oxidative stress, which are key areas of interest in functional food research.
3."an acidic environment suitable for probiotic stability" is incorrect – the sentence was revised L113
- section 2.1 is just basic nutritional information, not Physiochemical analysis- the section changed to Nutritional analysis L327
- n=2 is unacceptable We acknowledge that performing nutritional composition analyses in duplicate, rather than triplicate or higher replication, may be viewed as a limitation in terms of statistical robustness. Due to constraints in sample volume and analytical resources at the time of the study, the analyses were conducted in duplicate.
- no unit in table 2 -(Values are the mean of duplicate zone of inhibitions using well assays (mm) ± SD)
Reviewer 2 Report
Comments and Suggestions for Authors
- The main objective of the study was to evaluate a new drink obtained from whey with the addition of inulin, whey protein isolate and fruit juice concentrate that was fermented by probiotic bacteria.
- The topic of research undertaken by the authors is interesting. The manuscript presents the results of studies on the basic chemical composition of the drinks and its antimicrobial, antioxidant and angiotensin-converting enzyme (ACE) inhibitory activity.
- This manuscript contributes to the advancement of this field of knowledge compared to other published articles by using fermented undigested and digested samples.
- Notes in the methods section:
- Section 3.3.: This section is missing the origin of seven indicator bacteria that were given in Table 2, specifically those that do not have pure culture collection numbers, namely monocytogenes 33423, L. monocytogenes 33413, S. aureus 113, S. aureus Newman.
- Lines 17, 158 and Table 2 (line 175, column 1 and line 289: Please correct the spelling of the Latin name of the bacterium Salmonella Enteritidis, as it is an abbreviation of: Salmonella enterica subsp. enterica serovar Enteritidis.
- Line 170: Please correct the spelling of the Latin name of the bacterium Salmonella Typhimurium, because it is an abbreviation of: Salmonella enterica subsp. enterica serovar Typhimurium.
- Table 3: Correct the spelling of the abbreviation ( L. casei).
- The research results section is complete.
- The discussion is complete and properly discussed with the results of other authors.
- The conclution section is complete.
- References are adequate. 42 articles were used, 76% of which are from the last 10 years and 45% from last 5 years. This demonstrates the relevance of the research topic.
- There are two Table 2s in the manuscript. Please correct the table numbering and correct it in the text.
- Manuscript Title: The title should not contain abbreviations. The abbreviation ACE should have an expansion.
Author Response
Reviewer 2
- The main objective of the study was to evaluate a new drink obtained from whey with the addition of inulin, whey protein isolate and fruit juice concentrate that was fermented by probiotic bacteria.
- The topic of research undertaken by the authors is interesting. The manuscript presents the results of studies on the basic chemical composition of the drinks and its antimicrobial, antioxidant and angiotensin-converting enzyme (ACE) inhibitory activity.
- This manuscript contributes to the advancement of this field of knowledge compared to other published articles by using fermented undigested and digested samples.
- Notes in the methods section:
- Section 3.3.: This section is missing the origin of seven indicator bacteria that were given in Table 2, specifically those that do not have pure culture collection numbers, namely monocytogenes33423, monocytogenes 33413, S. aureus 113, S. aureus Newman.- table 3 was revised, origin/ characteristics of the strains were added
- Lines 17, 158 and Table 2 (line 175, column 1 and line 289: Please correct the spelling of the Latin name of the bacterium SalmonellaEnteritidis, as it is an abbreviation of: Salmonella enterica enterica serovar Enteritidis. - corrected
- Line 170: Please correct the spelling of the Latin name of the bacterium SalmonellaTyphimurium, because it is an abbreviation of: Salmonella enterica enterica serovar Typhimurium. -corrected
- Table 3: Correct the spelling of the abbreviation ( casei). -corrected
- The research results section is complete.
- The discussion is complete and properly discussed with the results of other authors.
- The conclution section is complete.
- References are adequate. 42 articles were used, 76% of which are from the last 10 years and 45% from last 5 years. This demonstrates the relevance of the research topic.
- There are two Table 2s in the manuscript. Please correct the table numbering and correct it in the text. - corrected
- Manuscript Title: The title should not contain abbreviations. The abbreviation ACE should have an expansion. – the title was revised
Reviewer 3 Report
Comments and Suggestions for Authors
This study aimed to develop a whey-based, fruit-flavored beverage with probiotics and a prebiotic, and to evaluate the antimicrobial, antioxidant, and ACE-inhibitory activities of the prepared products.
The topic is interesting, but the manuscript is not clearly written. Below are specific suggestions:
-
Abstract: “Results showed a significant increase in antioxidant activity following simulated gastrointestinal digestion.” Please clarify what is meant by “significant increase”—was it 1%, 5%, 50%, or several-fold? Add some quantitative detail to the abstract.
-
Please avoid abbreviations in the keywords.
-
Table 1: The reported potassium (2 g/L) and sodium (2.9 g/L) concentrations appear excessively high and could inhibit the growth of many probiotic strains. Please confirm these values, clarify the source of sodium, and discuss their potential impact on bacterial viability and fermentation kinetics.
-
What was the initial microbial count before fermentation (i.e., on “Day 0” in Fig. 1)?
-
Initial measurements on “Day 0” should also be included for all experiments where applicable.
-
I recommend renaming the samples according to the bacterial strain added and whether juice was used for fortification. This would make the manuscript easier to follow for the reader.
-
Fig. 3: How do you explain that the values in the digested samples are higher than in the undigested samples in all cases except sample no. 8?
-
Section 3.4: There is no need to describe the full cheese production process, as it is not directly relevant to your study.
-
Section 3.8: The description of antimicrobial evaluation is unclear. Please clarify whether the fermented beverage (including juice, inulin, and whey) was tested as a whole, or whether the cell-free supernatants of pure microbial cultures were tested separately. Also, specify the incubation temperature and duration. What was used as positive and negative control?
-
All equations must be numbered.
-
The equation in Section 3.10 is unclear and should be revised for clarity.
-
Was sensory analysis performed? If so, please include details; if not, this could be discussed as a limitation or future perspective.
-
Please expand the conclusions with more concrete takeaways from the study.
-
The reference list does not follow journal formatting guidelines. Latin names (e.g., bacterial species) must be italicized.
Author Response
Reviewer 3
This study aimed to develop a whey-based, fruit-flavored beverage with probiotics and a prebiotic, and to evaluate the antimicrobial, antioxidant, and ACE-inhibitory activities of the prepared products.
The topic is interesting, but the manuscript is not clearly written. Below are specific suggestions:
- Abstract: “Results showed a significant increase in antioxidant activity following simulated gastrointestinal digestion.” Please clarify what is meant by “significant increase”—was it 1%, 5%, 50%, or several-fold? Add some quantitative detail to the abstract. The sentence was revised. However, the extent of the increase in antioxidant activity after simulated gastrointestinal digestion varied substantially among the different samples, depending on their composition and formulation. Because of this variability, it is not possible to report a single representative percentage or fold increase. P19-20
Please avoid abbreviations in the keywords.- abbreviation was removed
Table 1: The reported potassium (2 g/L) and sodium (2.9 g/L) concentrations appear excessively high and could inhibit the growth of many probiotic strains. Please confirm these values, clarify the source of sodium, and discuss their potential impact on bacterial viability and fermentation kinetics. We have re-examined our measurements and confirm that the reported concentrations of potassium (2 g/L) and sodium (2.9 g/L) are accurate within experimental error. These values were derived using ICP method and the analysis was performed in duplicate. The elevated sodium and potassium levels are primarily attributable to the use of whey as the fermentation substrate. Whey naturally contains substantial concentrations of electrolytes, including sodium and potassium, depending on the origin of the milk, cheese processing conditions, and degree of concentration or demineralization applied prior to use. No additional sodium salts were intentionally added during formulation, other than those naturally present in the whey matrix (León-López, A., Pérez-Marroquín, X. A., Campos-Lozada, G., Campos-Montiel, R. G., & Aguirre-Álvarez, G. (2020). Characterization of whey-based fermented beverages supplemented with hydrolyzed collagen: Antioxidant activity and bioavailability. Foods, 9(8), 1106.). We recognize that these concentrations are relatively high and could potentially inhibit salt-sensitive probiotic strains. However, the strains used in this study (Lacticaseibacillus rhamnosus LGG, Lacticaseibacillus casei (431) and Lactobacillus helveticus (R0052) are known to possess moderate halotolerance and have previously been shown that were able to grow and survive in environments containing moderate salt concentrations especially up to 4% NaCl. (Rocha-Ramírez, L. M., Hernández-Chiñas, U., Moreno-Guerrero, S. S., Ramírez-Pacheco, A., & Eslava, C. A. (2021). Probiotic properties and immunomodulatory activity of Lactobacillus strains isolated from dairy products. Microorganisms, 9(4), 825.; Reale, A., Di Renzo, T., Rossi, F., Zotta, T., Iacumin, L., Preziuso, M., Parente, E., Sorrentino, E. and Coppola, R., 2015. Tolerance of Lactobacillus casei, Lactobacillus paracasei and Lactobacillus rhamnosus strains to stress factors encountered in food processing and in the gastro-intestinal tract. LWT-Food Science and Technology, 60(2), pp.721-728. Our results showed high sodium content did not affect the fermentation kinetics, and all strains showed consistent acidification profiles and maintained viability above 10⁷–10⁸ CFU/mL.
What was the initial microbial count before fermentation (i.e., on “Day 0” in Fig. 1)? Thanks for the comment. If you mean the microbial count (Total bacterial count) of the medium used for fermentation (i.e. whey) that was <10cfu/ml (unpublished results) as we have heat treated the medium (72°C/15sec) as it is stated in Figure 5.
- Initial measurements on “Day 0” should also be included for all experiments where applicable. Thanks for the comment. We have named the “Day 0 “samples as “Day 1”, as basically we determined the probiotic population after the addition of the probiotic culture and the time needed for fermentation to take place (18h).
- I recommend renaming the samples according to the bacterial strain added and whether juice was used for fortification. This would make the manuscript easier to follow for the reader. Table 4 in the text was added in order to give a full description of the samples used in the study.
- 3: How do you explain that the values in the digested samples are higher than in the undigested samples in all cases except sample no. 8? Thanks for the comment! Upon re-examination of the data, we identified an error in the reported results for Sample No. 8. Specifically, the values labeled as "undigested" were, in fact, the values for the "digested" sample, and vice versa. This mislabeling led to the inconsistency noted. We have corrected this error in the revised version of the manuscript to accurately reflect the data. With this correction, the trend of higher antioxidant activity in digested samples compared to undigested ones is consistent across all samples, including Sample No. 8.
- Section 3.4: There is no need to describe the full cheese production process, as it is not directly relevant to your study. The section was revised
- Section 3.8: The description of antimicrobial evaluation is unclear. Please clarify whether the fermented beverage (including juice, inulin, and whey) was tested as a whole, or whether the cell-free supernatants of pure microbial cultures were tested separately. Also, specify the incubation temperature and duration. What was used as positive and negative control? The section was revised. For the antimicrobial activity the fermented whey beverage as a whole was tested. In our study, uninoculated whey medium (processed under the same conditions as the test samples) was used as the negative control, ensuring that any observed inhibition zones were attributable to the strains used in this study and not to the medium components or processing conditions. For positive control, whey medium with a positive control strain isolated in previous studies was used.
- All equations must be numbered. Numbers added
- The equation in Section 3.10 is unclear and should be revised for clarity. The equation was revised.
- Was sensory analysis performed? If so, please include details; if not, this could be discussed as a limitation or future perspective. Sensory analysis was not performed as part of this study, as the primary focus was on the functional properties and shelf life stability of the fermented whey beverage. We acknowledge that sensory evaluation is an important aspect for assessing consumer acceptance and product development. Therefore, we have highlighted in the conclusion the absence of sensory analysis and proposing it as a key area for future research. L457-461
- Please expand the conclusions with more concrete takeaways from the study. The Conclusion section was revised.
- The reference list does not follow journal formatting guidelines. Latin names (e.g., bacterial species) must be italicized. The reference list was corrected
Round 2
Reviewer 1 Report
Comments and Suggestions for Authors
It has been well revised
Author Response
Thanks!
Reviewer 3 Report
Comments and Suggestions for Authors
I agree with the improvements of the paper.
Still some corrections are necessary:
- In Table 2, Sections 2.4 and 2.5, and Figure 4, points (decimal points) must be used instead of commas for numbers.
-
References should be described as follows: Author 1, A.B.; Author 2, C.D. Title of the article. Abbreviated Journal Name Year, Volume, page range.
Author Response
- In Table 2, Sections 2.4 and 2.5, and Figure 4, points (decimal points) must be used instead of commas for numbers.- corrected
- References should be described as follows: Author 1, A.B.; Author 2, C.D. Title of the article. Abbreviated Journal NameYear, Volume, page range - all references have been corrected.